# Allogenic Follicular Fosterage Technology: Problems, Progress and Potential

**DOI:** 10.3390/vetsci11060276

**Published:** 2024-06-17

**Authors:** Mingming Teng, Mengqi Zhao, Bo Mu, Anmin Lei

**Affiliations:** 1Guizhou Academy of Testing and Analysis, Guiyang 550013, China; 2Shaanxi Stem Cell Engineering and Technology Research Center, College of Veterinary Medicine, Northwest A&F University, Yangling 712100, China

**Keywords:** JIVET, oocyte maturation, follicle fosterage, embryonic development, ovulation disorders

## Abstract

**Simple Summary:**

Prepubertal female animals possess an extensive ovarian pool but are unable to ovulate naturally as adult females do. Through the process of transferring oocytes from prepubertal females into the dominant follicles of adult females, a significant number of oocytes can be obtained while minimizing the adverse effects of the in vitro environment. This innovative technology offers an alternative approach to embryo production, independent of complex laboratory equipment. However, the reproducibility and efficiency of this technology are still insufficient for widespread clinical applications. This review shares the latest progress of this technology, outlines the current limitations, and summarizes the factors that affect the success rate of this technology.

**Abstract:**

The allogeneic follicular fosterage (AFF) technique transfers cumulus–oocyte complexes (COCs) from pubertal female animals to the dominant follicles of adult female animals for further development, allowing the COCs to further develop in a completely in vivo environment. This article reviews the history of AFF and JIVET and their effects on oocyte and embryo development as well as freezing resistance. Improving the efficiency and reproducibility of AFF technology is crucial to its clinical application. This article discusses factors that affect the success rate of AFF, including differences in specific technical procedures and differences between pubertal and adult follicles. Designing standardized procedures and details to improve the synchronization of donor COCs and recipient follicle maturity and reducing the damage to COCs caused by follicular aspiration may be the direction for improving the success rate of AFF in the future.

## 1. Introduction

In vitro embryo production (IVEP) technology has had a momentous impact on livestock production, with a significant increase in its application in recent years. In vitro produced embryos still differ in many ways from embryos derived from in vivo sources [1]. Compared to embryos developed in vivo, the survival rate of embryos produced in vitro is lower [2], which is a major obstacle to further developing this embryo technology. Even if the embryos produced in vitro can develop to the blastocyst stage, they are not as resilient to freezing [3], have a different ultrastructure [4], microvilli [3], lipid content [5], and gene expression patterns [6], and have a higher incidence of chromosomal abnormalities [7] compared to embryos developed in vivo. In order to overcome the limitations of IVEP, Fleming and his colleagues first introduced a technique in 1985 that allowed the transfer of oocytes from baboons and cattle into pre–ovulatory follicles of adult mares (intrafollicular oocyte transfer, IFOT) [8]. IFOT includes autologous or allogeneic IFOT. Allogeneic transfer is also known as allogenic follicular fosterage (AFF). Autologous transfer entails transferring oocytes from one’s own follicles into one’s pre–ovulatory follicles, thus avoiding the need to retrieve undesired embryos from follicles [9]. AFF is the process of fostering allogeneic COCs derived from young female animals into the follicles of adult female animals for development. From the perspective of clinical application, AFF has multiple advantages. Oocytes for allogenic transplantation can be transplanted immediately or stored overnight before transplantation, which may be suitable for situations where oocytes are transported to central facilities. Moreover, AFF can avoid excessive manipulation of more valuable female donors [10]. When donor female animals have reproductive pathologies and cannot provide embryos, AFF can also provide a new avenue. It is noteworthy that in a study investigating the application of AFF in horses, the ovulation rate and the yield of additional embryos achieved through AFF were significantly greater than those achieved through autologous transfer [11]. Therefore, AFF may be the first choice for future clinical applications.

In recent years, AFF technology has successfully demonstrated blastocyst procurement and viable offspring production across multiple animal models [11,12,13,14,15]. Its potential implementation in clinical settings promises to significantly enhance and expand the scope of JIVET technology. This is primarily due to its ability to generate a substantial quantity of embryos within the in vivo system, thereby minimizing the adverse effects typically associated with the in vitro environment on oocyte development. Additionally, AFF technology eliminates the need for elaborate laboratory infrastructure, reducing economic costs and simultaneously improving breeding efficiency. Furthermore, it holds promise in elucidating the underlying mechanisms of follicular ovulation disorders in juvenile female animals and providing deeper insights into the influence of culture environments on oocyte and embryo development. However, despite its potential, there is currently a lack of comprehensive reviews on AFF technology, hindering its broader recognition and adoption among researchers. To address this gap, the present review aims to provide a detailed exposition of the historical background, methodologies, and advancements of AFF technology. Given the current limitations regarding repeatability, this review also delves into the factors impeding its success rate. This analysis paves the way for the development of more standardized, efficient, and reproducible technical protocols in the foreseeable future.

## 2. The History of JIVET and AFF and Their Impact on Oocyte and Embryonic Development

In large adult animals, most of the oocytes used for commercial embryo production are collected through ultrasound–guided ovum pick–up (OPU). However, for species that are too small to be collected through OPU, such as sheep and goats, the development of LOPU in the early 1990s has made it possible to collect oocytes more safely [16]. Compared to OPU, LOPU can better observe the ovary and reduce the risk of ovarian damage. More importantly, LOPU technology enables the safe collection of oocytes from the follicles of young female animals before sexual maturity, which, in turn, has led to the development of juvenile in vitro embryo transfer (JIVET). JIVET technology can obtain a large number of oocytes from young female animals in vivo, which can produce blastocysts through in vitro fertilization and then transfer them into the uterus of adult recipient females to produce offspring. Using this method, multiple offspring of donor animals can be born before reaching sexual maturity. JIVET has outstanding advantages. Firstly, compared to adult females, young female animals have a very large ovarian pool, are very sensitive to reproductive hormones, and have fewer follicular atresia, which can consistently produce a large number of COCs. Moreover, early breeding of high–quality female livestock can shorten the generation interval and increase the genetic gain rate. Taking buffalo as an example, if a calf receives LOPU at two months old, the offspring will be born around one year old in the donor animal, effectively shortening the generation interval by two years. From a basic research perspective, animals before sexual maturity are a good model for studying the mechanism of ovulation disorders. Oocytes of prepubertal young animals enter metaphase II (MII) through meiosis at a similar speed to adult animal oocytes, and then can be fertilized, developed, and produce viable blastocysts [17]. They can then be transferred to adult recipients and eventually produce live offspring [18]. The oocyte production of juvenile animals is high, usually higher than that of adult animals [19]. However, although their fertilization rate and cleavage rate are similar to those of adult animals, their blastocyst rate is always lower than that of adult animals [19,20]. Moreover, although there are many follicular developments on the ovary of juvenile animals after superovulation, they cannot complete the normal ovulation process like adult animals. The reason for the impaired developmental ability of juvenile animal oocytes may be a combination of multiple factors [21]. For example, the immature hypothalamic–pituitary–ovarian axis in prepubertal animals may lead to abnormal signaling and steroid production in ovarian follicles [22]. Other studies have found that the LH concentration and estradiol concentration in the follicular fluid of juvenile animals are about half of those of adult animals, and such a follicular environment may affect the metabolism of oocytes themselves, as well as the signaling between oocytes and granulosa cells, ultimately leading to the inability of oocytes to achieve full developmental ability [23].

In addition, JIVET technology requires IVEP, and embryos produced in vitro are still different from embryos of in vivo origin in many aspects [1]. Compared with embryos developed in vivo, the survival rate of embryos produced in vitro is lower, which is a major obstacle to the further implementation of this embryo technology. Taking cattle as an example, generally only 30–40% of oocytes from slaughterhouse–derived ovaries have sufficient ability to develop into blastocysts after in vitro culture, while about 90% of oocytes excreted in vivo are fertilized after insemination, and most of them develop into blastocysts [24]. Even if the embryos produced in vitro can reach the blastocyst stage, they are inferior to those produced in vivo in terms of freezing resistance, ultrastructure, microvilli, lipid content, gene expression, and the incidence of chromosomal abnormalities. In addition, various studies have shown that the cultivation of embryos in vitro not only leads to changes in the expression of transcripts related to metabolism and growth, but also affects the development of pregnancy and fetuses after transplantation [25,26,27]. It has been reported that the addition of serum to the embryo culture medium can lead to significant changes in the transcriptome of embryos [28]. Earlier, it was reported that the addition of serum containing fatty acids to the embryo culture medium can affect the quality of embryos [29] and the survival of embryos after freezing and thawing [30]. Moreover, due to the poor freezing resistance of embryos produced in vitro, more than 90% of the embryos produced in vitro for about 300,000 cattle transplanted each year are freshly transplanted, reducing the flexibility associated with surgery [31,32].

In order to overcome the limitations of in vitro embryo culture, various methods for producing embryos in vivo or in vitro have been studied. As the critical role of the fallopian tube in supporting early embryo development has been widely accepted, mouse fallopian tubes [33], rabbit fallopian tubes [34], and sheep fallopian tubes have been widely used for in situ bovine embryo development [25,31,35]. Besenfelder and colleagues established a minimally invasive endoscopic technique that allows access to the bovine fallopian tubes of living animals and combines transplantation and flushing to culture embryos in vivo [36,37]. However, the intrafallopian transfer of embryos requires high levels of skill, which hinders its widespread use. As an alternative, Fleming and colleagues first introduced a technique in 1985 that can transfer oocytes into the pre–ovulatory follicles of fertilized recipient mares [8]. Mares can provide a good model for this procedure because their pre–ovulatory follicles are large, approximately 35–45 mm in diameter, and covered with a thick white membrane, which can resist damage during puncture. It is noteworthy that in the study on the oocyte transfer of horses, the ovulation rate of autologous transfer was 69%, while that of AFF was 91%. Among the horses with autologous transfer, 15% of the mares produced embryos, while 64% of the AFF mares produced embryos. In 13 cases of autologous transfer, no additional embryos were produced, while in 11 cases of AFF, 4 out of 11 cases produced additional embryos that were verified by kinship testing. There was no significant difference in the diameter of autologous and AFF embryos [11]. Therefore, AFF may be the first choice for future clinical applications. In mares, AFF as a method for producing foals from isolated oocytes also has clinical potential, as standard in vitro fertilization has not been successful in this species [38]. The alternative method of intracytoplasmic sperm injection (ICSI) is expensive and only available in some places. In addition, mares are single ovulators, although like women, they may sometimes ovulate two or more follicles, but the probability is low, and effective superovulation methods have not yet been developed for this species. In addition to horses, Kassens and his team were able to achieve a significant milestone by utilizing the AFF technology to successfully obtain bovine AFF blastocysts, leading to the birth of healthy calves for the first time [13].

AFF technology allows the development of oocytes and embryos in juvenile female animals to be completed in the in vivo environment, thus reducing the negative effects of in vitro environments, such as oxidative stress, on oocyte and embryo development. To investigate whether AFF technology promotes oocyte and embryo development as expected, Hoelker and colleagues transferred a total of 791 immature bovine oocytes into the pre–ovulatory follicles of 16 synchronized cows. They recovered 306 additional oocytes/embryos at a 38.5% efficiency. Of all the recovered oocytes/embryos, 40.1% had developed to the morula and/or blastocyst stage, indicating an overall efficiency of 17.3% based on all transferred oocytes. Importantly, when compared to in vitro cultured control embryos from the same batch of slaughterhouse ovaries, AFF embryos exhibited significantly higher development rates to the morula and/or blastocyst stage on Day 7, at 40.1% and 29.3%, respectively [14]. Therefore, the follicular environment likely has a beneficial impact on the intrinsic quality of fertilized embryos during development and their subsequent rate of progression to the blastocyst stage. The study also revealed that transferring immature oocytes into pre–ovulatory follicles in recipients led to higher embryo recovery rates and further development compared to mature oocytes, possibly due to better synchronization between ovulation time and oocyte maturation time. This finding aligns with previous research conducted on horses, where a recent study showed that the overall success rate of transferring immature oocytes into female horses was generally higher [9]. Simões and colleagues compared the embryo production rates of bovine oocytes after in vitro maturation (IVM) or after in vivo maturation for 20 or 28 h, and they found no significant difference in the embryo production rates of each group. However, interestingly, the fertilization rate of oocytes matured in vivo for 20 h was higher than that of IVM oocytes [39]. Andino and colleagues reported that there was no significant difference in the maturation rate of horse oocytes between AFF technology and in vitro culture systems, with 56% and 49%, respectively. After ICSI, the blastocyst rates were 25% and 18%, respectively [11].

However, the reproducibility of AFF technology in enhancing the developmental potential of oocytes and embryos remains limited. In a study conducted by Kassens and colleagues, upon transferring 1646 in vitro matured bovine oocytes into 28 follicular recipients, 583 embryos were recovered via uterine flushing, representing a recovery rate of 35.2%. Notably, when compared to embryos cultured in vitro, the cleavage rate of AFF embryos was significantly reduced, standing at 63.2% compared to 88.8% for in vitro embryos. Furthermore, the blastocyst development rate of AFF embryos was also comparatively lower, at 8.0% versus 36.5% for in vitro embryos [13].

Due to the significant differences in the success rate of AFF obtained in different studies, the design and specific details of AFF research vary, which may affect the final success rate. For example, the success rate of transferring immature COCs is higher than that of transferring mature COCs, and the success rate of transferring COCs with more layers of granulosa cells is higher than that of COCs with fewer layers of granulosa cells. Therefore, improving the synchronization of COCs and recipient follicle maturation and reducing physical damage to COCs during follicular aspiration may increase the final success rate of AFF. This paper will subsequently delve into various factors that influence the success rate of AFF technology.

## 3. AFF Steps and Distinguishing AFF COCs from Originals

The main steps of AFF technology are shown in Figure 1: (1) Selecting young female donors (taking lambs aged 5–8 weeks and weighing about 15 kg in Figure 1 as an example); (2) Using hormones and other methods to induce superovulation in young animals, and then obtaining a large number of COCs through laparoscopic ovum pick–up (LOPU); (3) Synchronizing estrus in adult female animals, and confirming the correlation time between ovulation and withdrawal of the embolus through rectal ultrasound monitoring; (4) Transferring the collected COCs to the pre–ovulatory follicles of adult recipient female animals; (5) Mating or artificial insemination; (6) On the 7th–8th day after embryo development, performing uterine flushing to recover the embryos, and obtaining offspring following transplantation.

In the study conducted by Andino and colleagues, they differentiated the native COC from the AFF COCs based on the characteristic large, yellow, mucoid expanded cumulus cells of the latter. AFF–derived COCs had smaller and less cellular cumulus masses because most of their cumulus cells were stripped during the initial recovery from their native follicles. When the cumulus cells of AFF–derived COCs expanded, they appeared to have a clear rather than yellow intercellular matrix and were less mucoid in appearance [11]. To further accurately determine the origin of the embryos, the recovered embryos were carefully bisected, and the resulting cells were then submitted for parentage testing [11]. The offspring obtained through AFF can be confirmed using genotype analysis [13].

## 4. The Impact of AFF on the Antifreeze Capacity of Oocytes and Embryos

To date, the potential of cryopreserved oocytes to develop into blastocysts after warming has been disappointing in most livestock species. The high lipid content in the cytoplasm and the specific phospholipid composition of the cell membrane are frequently identified as the primary factors behind this limitation [40,41,42]. Research has demonstrated that IVM leads to an increased accumulation of lipid droplets, subsequently elevating the lipid content in future embryos. Intriguingly, Faria and colleagues observed that the average lipid area in oocytes matured in vivo, as well as those undergoing AFF, was similar to that in immature oocytes. However, this area was notably smaller compared to oocytes matured in vitro. Remarkably, even when oocytes were cultured in a medium containing bovine serum albumin (BSA), their lipid accumulation remained higher than that of oocytes matured in vivo or through AFF. Transmission electron microscopy (TEM) analysis of the oocytes’ ultrastructure corroborated the findings observed through confocal microscopy, indicating that in vitro matured oocytes tend to contain a higher concentration of lipid droplets than those matured in vivo or through AFF. This observation suggests that AFF technology may potentially enhance the cryotolerance of oocytes by reducing their lipid content [43]. In another study, researchers observed a significant decrease in lipid content and a marked increase in freezing tolerance in embryos derived from the AFF technique, when compared to embryos cultured in vitro in CR1aa medium supplemented with estrus bovine serum (ECS). However, when these embryos were cultured in SOFaa medium enriched with fatty acidfree bovine serum albumin (BSA–FFA), the differences in lipid content and freezing tolerance between the AFF–derived and in vitro–produced embryos were no longer significant [13].

Could the application of AFF technology potentially reverse the detrimental effects inflicted by vitrification cryopreservation on oocytes? Sprícigo and colleagues explored this question by conducting an experiment that entailed resuscitating vitrified immature oocytes and maturing them either through in vitro culture or utilizing AFF technology. Their findings revealed that the blastocyst rate for oocytes that underwent vitrification cryopreservation and subsequent AFF treatment was 0%, significantly trailing the 6.3% blastocyst rate achieved through in vitro culture [15]. The underlying reasons for this outcome may be multifaceted, ranging from potential oocyte loss during the injection process to the possible loss of cumulus cells during transfer. Notably, vitrification not only threatens the survival rate of oocytes but also impacts cumulus cells in some manner, resulting in compromised cumulus expansion, a crucial process for facilitating the capture and transportation of oocytes by the ciliated epithelial cells of the fallopian tube umbrella to the site of fertilization [44]. Consequently, researchers propose that the damage inflicted on oocytes by vitrification cryopreservation may be intractable, posing a significant challenge even when attempting to mature them within follicles post–warming and resuscitation.

## 5. Factors That Can Be Relevant for a Successful Transfer

With the knowledge that AFF technology is indeed effective, as evidenced by the birth of healthy calves after embryo transfer, the current challenge is to improve efficiency. Standardizing AFF technology is the key to improving efficiency, but the factors that affect its success rate are not fully understood, and it is likely to be a combination of multiple factors. Currently, factors reported in the research include the size of the transfer needle, the quantity and quality of the oocytes, the synchronization of the ovulation of the transferred oocytes and the foster follicles, differences between species and individuals, the proficiency of the surgical personnel, and so on.

### 5.1. The Size of the Transplant Needle

The sizes of follicles and oocytes vary among different species, and transplant needles that are too large or too small may have a negative impact on the follicle transplantation of oocytes. It is necessary to select a needle with an appropriate diameter based on the species of different animals during transplantation. If the diameter of the syringe needle used is too large (such as 15–18 G), it may cause follicular trauma or leakage of fluid within the follicle and loss of COCs [38]. For example, in the study of Hinrichs and colleagues on the AFF of horses, ovarian bleeding was observed in 3 of the 20 horses [10]. A needle with a too small a diameter (such as 30 G) can damage the structure of the injected COCs and lead to the loss of granulosa cells. The integrity of COCs is an important parameter, as the close coupling between oocytes and cumulus cells is crucial for developmental capacity [45]. In the study of Falchi and colleagues on AFF in sheep, they found that after using a 27 G needle, liquid reflux was observed in punctured follicles, and the recovery rate of oocytes after aspiration was 38%. After 14 injections using a 30 G needle, the recovery rate increased to 68%, and no reflux was observed, but all recovered oocytes were partially or completely denuded. Subsequently, another 25 injections were performed using a 28 G needle, resulting in a recovery rate of 75.5%, and a better preservation of the integrity of COCs was observed. Therefore, 28 G was considered to be the most suitable needle diameter for maintaining the integrity of sheep COCs and preventing reflux [12]. In the study of Martinez and colleagues on AFF in horses, after using a smaller gauge needle (20 G) for injection compared with using a larger gauge needle, although the incidence of hemorrhagic anovulatory follicles (HAFs) formation did not decrease significantly [11] (11.5% [11] vs. 15.8% [10] vs. 20% [9]), and the proportion of mares providing AFF embryos was not significantly higher, using a smaller gauge needle may help to reduce damage to the follicles, thereby increasing the recovery rate and maturation rate of AFF oocytes and promoting the production of AFF embryos.

### 5.2. Number and Quality of Transplanted Oocytes

The number of transplanted oocytes may also affect the efficiency of transplantation. Research by Simões and colleagues on dairy cows has shown that transplanting excessive oocytes into dominant follicles may affect the release of oocytes and the capture of the fimbria of the fallopian tube [39]. However, transplanting too few oocytes may result in fewer available embryos, reducing the efficiency of AFF and increasing costs. Therefore, according to different species and different maturities of oocytes, transplanting the appropriate number of oocytes is one of the important factors to improve the efficiency of AFF. Sprícigo and colleagues examined the follicles of adult cattle before AFF, using only recipients with a single dominant follicle. Then, they injected 10 to 25 oocytes into the dominant follicles of each recipient [15]. Simões and colleagues injected 25–35 COCs into each dominant follicle in dairy cows [39]. Hoelker and colleagues transplanted 50 COCs from slaughterhouses into each dominant follicle in cattle [14], while Kassens and colleagues used a higher number, namely 60 COCs from cattle [13]. Given the small size of the dominant follicles in sheep, Falchi and colleagues injected 20–30 oocytes into each dominant follicle in sheep [12].

Sprícigo and colleagues conducted two experiments using the same number of oocytes and the same diameter of recipient follicles, and found that the extra blastocyst rate obtained using OPU was 5.6%, while the extra blastocyst rate obtained using COCs obtained from slaughterhouses was 12.9%. They believed that the reason for this difference was that slaughterhouse ovaries provide a large number of COCs that can undergo more rigorous screening, resulting in higher–quality oocytes [15].

### 5.3. The Size of Foster Follicles

Kassens and colleagues found that by transplanting oocytes into pre–ovulatory follicles of different diameters in cattle, they could produce more extra embryos (34.3 vs. 7.3) and more extra morula and blastocysts (8.3 vs. 0.8) when using follicles with a diameter of 13–14 mm compared to those with a diameter of 9–10 mm [13]. Falchi and colleagues also reached similar conclusions in the pre–ovulatory follicles of sheep; that is, the follicular diameter is positively correlated with the success rate of transplantation [12]. The composition of follicular fluid varies with follicular size, with larger follicles containing more glucose and cholesterol [46]. The gradual increase in follicle size near ovulation may be consistent with the optimal fluid composition of hormones and metabolites, suitable for the nuclear and cytoplasmic maturation, fertilization, and embryo development of fostered oocytes [12]. However, another study showed that there was no significant difference in the cleavage rate of embryos after transferring oocytes into recipient follicles of different diameters, suggesting that the recovery rate of embryos is more related to the transfer technique rather than the diameter of the fostered follicle [14].

### 5.4. Synchronization of the Maturation of Transplanted Oocytes and Recipient Follicles

Research has found that embryos produced by transplanting pre–ovulatory follicles after in vitro maturation culture exhibit lower cleavage rates compared to embryos produced entirely through in vitro culture systems, which may be related to poor synchronization between follicular and oocyte maturation [13]. After undergoing in vitro maturation culture and subsequent transplantation into pre–ovulatory follicles, embryos have been found to exhibit lower cleavage rates compared to those entirely cultivated in vitro. This disparity may stem from the inadequate synchronization between the maturity of the follicles and oocytes [47]. Research has found that the maturation rate of cow oocytes that matured in vivo for 28.3 h is lower than that of oocytes that matured in vivo for 19.8 h, indicating that extending the maturation time can reduce the maturation rate of oocytes [39].

Therefore, the timing of the administration of ovulation–inducing drugs to control the time of follicular ovulation after AFF may affect the development of transplanted oocytes. The most appropriate timing for AFF is still unclear. The pre–ovulatory follicles in horses ovulate 36 to 42 h after the administration of ovulation–inducing drugs [48]. For oocytes recovered from immature follicles and subjected to IVM, the blastocyst rate of ICSI reaches its highest level after 24 to 36 h of IVM [49]. If the oocytes are stored at room temperature overnight before maturation begins, the maturation time required for oocyte development may be shorter [50]. Deleuze and colleagues used equine pituitary gonadotropins to promote ovulation during oocyte transplantation [9]. In the study by Hinrichs and DiGiorgio, hCG was used to promote ovulation in horses 16 h before oocyte transplantation [10]. In order to provide sufficient maturation time for transplanted oocytes, Sprícigo and colleagues recommended that AFF should be performed at least 18 h before ovulation in cattle [15]. In the study by Falchi and colleagues, the time between the removal of the progesterone device in sheep and ovulation was 60–72 h. Considering that sheep oocytes need to remain in the follicle for at least 20 h before ovulation to achieve maturation, they transplanted the oocytes into follicles with a size of 5–7 mm 40 h after device removal [12].

In the future, an effective strategy to enhance the synchronization between the maturation of transplanted oocytes and recipient follicles is to transfer oocytes matured in vitro closer to the ovulation time of the recipient follicles. This approach can minimize the aging process of the oocytes, ultimately leading to higher developmental rates and an overall boost in the efficiency of AFF. Utilizing ultrasound to evaluate follicular diameter and vascularization provides valuable insights into determining the precise ovulation timing of the recipient follicles [15]. Moreover, studies have revealed that transplanting immature oocytes yields a greater number of additional embryos with a cleavage rate of 83.2%, suggesting that these oocytes exhibit greater synchronization with the recipient’s dominant follicles compared to oocytes matured externally [14]. Therefore, injecting oocytes that are partially matured (less than 16 h) or immature into the recipient follicles before ovulation could be a viable option.

### 5.5. Species and Individual Differences

Species differences refer to the disparities in ovarian structure and size among different animal species. For instance, horses exhibit pre–ovulatory follicles with diameters ranging from approximately 35 to 45 mm, encased in a thick white membrane that resists damage during puncturing procedures. This characteristic potentially mitigates the negative impact of surgical interventions on horse AFF. However, mares are naturally monovulatory creatures, and although they may occasionally ovulate two or more oocytes, the probability is minimal. Furthermore, the lack of effective superovulation methods for this species further compounds the challenges associated with performing AFF in mares.

Individual differences are a pervasive phenomenon in animal experiments, and they are equally prevalent in the realm of AFF research. In a study conducted by Hoelker and colleagues focusing on AFF in cattle, they utilized follicles from 16 cows for AFF. The results revealed significant variations in the total embryo recovery rate, with 19.1 ± 13.7 embryos recovered from approximately 50 oocytes transferred per pre–ovulatory follicle [14]. Specifically, the average count of extra cleavage embryos per pre–ovulatory follicle stood at 15.6 ± 11.3, while the average number of developing embryos, including morula and blastocysts, was 8.6 ± 7.2. Strikingly, four cows (25%) yielded minimal or no embryos, despite a total recovery rate spanning 21.3% to 74.0% in the remaining recipients. Interestingly, there was no apparent correlation with follicle diameter. Furthermore, among these 4 cows (25%), the development rate of morula and/or blastocyst stages was nil, whereas the remaining 12 cows exhibited a developmental embryo proportion ranging from 30.0% to 67.6%. Similar phenomena were also found in the study by Kassens and colleagues, where they recovered 20.8 ± 17.7 embryos from approximately 60 oocytes transferred from each recipient follicle, with an average number of extra cleavage embryos of 13.8 ± 13.0 [13].

### 5.6. Other Factors

In addition to the aforementioned factors, the procedural details of the experiment, the surgical proficiency of the operators, the age of the donors and recipients, and other variables can also influence the ultimate success rate of AFF procedures.

The standardization of the AFF technical process holds paramount importance in ensuring its success rate. However, there remains a lack of clarity regarding the crucial factors and mechanisms that contribute to the success of AFF, indicating that further advancements and optimization are necessary. Designing an efficient and reproducible protocol, along with developing surgical methods that are simpler and less invasive, is imperative for the widespread adoption and practical application of this technology.

## 6. Factors Influencing Oocyte Developmental Differences between Prepubertal and Adult Female Animals

One of the significant challenges in addressing the compromised developmental potential of prepubertal oocytes stems from the incomplete understanding of the underlying causes, which are likely attributed to a multifaceted array of factors. For instance, the immaturity of the hypothalamic–pituitary–ovarian (HPO) axis in prepubertal animals may result in deficiencies in signal transduction and steroidogenesis within ovarian follicles. Consequently, a suboptimal follicular microenvironment can adversely impact oocyte metabolism or intercellular communication between oocytes and granulosa cells, ultimately hindering oocytes from reaching their full developmental potential. Elucidating the factors that contribute to the disparities in developmental capacity between juvenile and adult animal oocytes is crucial for elucidating the principles of JIVET and AFF techniques, thereby enhancing their efficiency.

### 6.1. Follicular Microenvironment

The diminished developmental potential of calf oocytes is often attributed to deficiencies in the follicular environment prior to their retrieval [23]. Therefore, a comprehensive understanding of the disparities in the follicular structure and its follicular fluid between juvenile and adult animals is crucial. Specifically, the luteinizing hormone (LH) concentration in calf follicular fluid is approximately half of that observed in adult cow follicular fluid (2.0 ± 0.2 ng/mL vs. 4.0 ± 0.3 ng/mL) [51]. This finding aligns with the reduced LH concentration in the plasma of younger animals [52]. Notably, despite the absence of LH receptors in oocytes, alterations in LH concentration may indirectly impact their development by modulating steroidogenesis and androgen production in granulosa and theca cells [53].

Furthermore, disruptions in estrogen production can influence the transcription of genes regulated by estrogen response elements. Additionally, impaired androgen metabolism can also compromise fertility, as evidenced by the suboptimal fertility exhibited by androgen receptor knockout mice [54]. Similarly, the estradiol content in calf follicular fluid is approximately half of that in adult cow follicular fluid (6.3 ± 2.1 ng/mL vs. 12.7 ± 5.5 ng/mL) [51].

Collectively, these disparities in the follicular microenvironment may adversely affect the developmental capacity of oocytes in juvenile animals, partially explaining the lower IVEP results observed in calves [23]. This rationale supports the potential of AFF to enhance oocyte developmental capacity, as the transfer of oocytes from juvenile animals into adult follicles provides them with a more favorable environment for development.

### 6.2. HPO Axis and Follicular Diameter

The HPO axis plays a pivotal role in regulating the estrous cycle and subsequent fertility in juvenile animals. In juvenile animals, the immaturity of the HPO axis can result in defects in signal transduction and steroidogenesis within ovarian follicles. Compared to adult cows, the relative mRNA abundance of FSH receptors in the granulosa cells of prepubertal animals is significantly lower, potentially explaining the smaller average size of the follicles and subsequently reduced developmental potential of the oocytes [22]. The molecular changes occurring during follicular and oocyte growth, involving molecules synthesized within oocytes or imported from granulosa cells, are crucial for acquiring oocyte developmental potential. Numerous studies have demonstrated a positive correlation between follicular diameter and oocyte developmental potential in various species, including sheep [55], goats [56], cattle [57], buffalo [58], and pigs [59]. For instance, in adult cattle, oocytes derived from follicles with diameters ranging from 2 to 6 mm exhibited an average blastocyst rate of 34.3%, while oocytes from follicles with diameters greater than 6 mm had an average blastocyst rate of 65.9% [60]. A similar pattern was observed in adult buffalo, wherein oocytes from follicles with diameters <3 mm had a blastocyst rate of 2.4 ± 1.5%, while oocytes from follicles greater than 8 mm exhibited a blastocyst rate of 16.9 ± 1.7% [61]. In prepubertal animals, a similar trend was noted, with the blastocyst rate increasing from 6.8% to 13.8% for oocytes derived from small (<5 mm) and large (≥5 mm) follicles, respectively, in calves [62]. These findings potentially elucidate the influence of foster follicle size on oocyte development in AFF, suggesting a positive correlation between foster follicle diameter and transfer success rate.

### 6.3. Interaction between Oocytes and Granulosa Cells

The capability of oocytes relies on intercellular communication during the growth and development of ovarian follicles, which is regulated by endocrine, paracrine, and autocrine factors [63]. While direct intercellular connections are mediated through gap junctions and transzonal projections (TZPs) [64], indirect intercellular communication can occur via extracellular vesicles (EVs) released into follicular fluid [65]. These pathways collectively facilitate bidirectional communication, signal transmission, and molecular transportation between oocytes, granulosa cells, and theca cells [65]. As follicles and oocytes grow, their increasing transcriptional activity progressively and sequentially enhances the developmental capability of oocytes [60]. This is crucial as oocytes from prepubertal animals are smaller and have a thinner zona pellucida compared to oocytes from adult animals, even when follicles of similar sizes are derived from both prepubertal and adult animals. For instance, the average diameter of oocytes from calves is 118.04 ± 1.15 μm, whereas the average diameter of oocytes from mature cows is 122.83 ± 0.74 μm [66]. Since small changes in diameter represent larger changes in volume, these minor differences in diameter may have significant impacts on developmental capability. Consequently, the ability of bovine oocytes to reach MII during IVM is positively correlated with their diameter [67]. In addition to diameter, some cytoplasmic differences have also been observed between oocytes from prepubertal and adult animals. For example, oocytes from adult cows contain more lipid droplets in their cytoplasm before and after IVM compared to oocytes from young cows [23]. Other differences include incomplete cytoplasmic maturation, alterations in gene expression and protein synthesis, and metabolic defects in oocytes from younger animals [57,68].

Recently, researchers have begun to delve into the close relationship between oocytes and cumulus cells to better define the role of TZPs in cross–zonal projection [69,70]. Although more research is needed to determine how the physiology, distribution, and retraction of TZPs affect the IVEP efficiency of prepubertal and adult oocytes, it is known that TZPs can promote communication and the transport of essential molecules between granulosa cells and oocytes [71]. Despite the differences observed in the organization of TZPs in COCs from lambs compared to adult ewes, the impact on embryo development remains unclear [72]. In addition to intercellular communication through TZPs, the role of EVs in cell–to–cell communication within follicles has also attracted particular attention [65]. EVs are small lipid bilayer particles secreted by cells into the extracellular space, which subsequently spread and act on secondary target cells, transporting various molecules including proteins, lipids, messenger RNA (mRNA), and microRNA (miRNA) [73]. Since their first discovery in the follicular fluid of mares [74], they have subsequently been described in the follicular fluid of cattle [75] and pigs [76] and have been shown to play multiple roles within follicles, including granulosa cell proliferation and cumulus expansion [77]. It is noteworthy that studies have found differences in the EV and miRNA profiles when comparing follicular fluids from follicles of different sizes and young versus old animals [77,78]. For example, da Silveira found significant differences in the number and pattern of miRNAs when comparing the follicular fluids of mares aged 3–13 years with those aged over 20 years [79]. Others have also found similar results when comparing women aged under 31 years with those over 38 years [80]. Understanding these differences and patterns can help us to further understand the mechanisms of AFF and improve the efficiency of AFF and JIVET.

## 7. Potential Negative Factors Influencing the Success Rate of AFF

During the process of collecting COCs from follicles in prepubertal female animals through follicular aspiration, the collected oocytes are only encapsulated by a few layers of granulosa cells due to the immature nature of these follicles. The lack of granulosa cells may affect the adhesion of COCs to the follicular wall, causing them to float in the follicular fluid, which is not conducive to the transfer of signals from the follicular environment to the oocytes, resulting in some transferred oocytes failing to ovulate normally. In a study on horses using the AFF technique, experimental results showed that the recovered oocytes were all present in the initial follicular aspiration fluid, not in the follicular flushing fluid [11]. During ovulation in mares, most of the follicular fluid seems to be lost into the peritoneum [81]. Therefore, transferred free–floating COCs may be lost into the peritoneum along with the initial fluid. Similarly, in another experiment on cattle, about 60% to 65% of unsuccessfully recovered oocytes and embryos were also suspected to be oocytes that did not enter the follicle after injection, or that remained in the follicle or entered the pelvic cavity after ovulation, resulting in their failure to reach the fallopian tubes during ovulation or to enter the uterine cavity for unknown reasons after several days [14]. In addition, the reduction in the number of granulosa cells may also affect the capture of oocytes by the fallopian tubes during ovulation. Alternatively, this may be due to the strong adhesion of granulosa cells in transplanted COCs, which fail to separate properly or adhere to the wrong location, resulting in oocytes being unable to be released from the follicle during ovulation [11]. During the maturation of oocytes, hyaluronic acid is produced, which enhances the adhesion of granulosa cells [82] and inhibits the release of oocytes [83]. Therefore, the excessive hyaluronic acid resulting from transplanting a large number of oocytes into dominant follicles through AFF technology may affect the release process of oocytes, thereby reducing the ovulation rate. The selection of an appropriate number of oocytes for transfer, tailored to the specific follicular sizes of various species, represents a crucial factor in enhancing the efficacy of AFF technology.

On the other hand, it has been reported that follicular aspiration before ovulation can lead to a smaller luteal volume and lower progesterone levels [84]. The follicular diameter at ovulation and the duration of the proestrus period are considered important factors affecting pregnancy rates [85,86,87]. The delay in the increase in progesterone levels after ovulation is associated with a delay in pregnancy development [88]. Although ultrasound studies have shown that the size and blood supply of the corpus luteum after AFF remain unchanged [12,15], these studies did not measure progesterone levels, so whether follicular aspiration has a negative impact on the development of AFF embryos still needs further research.

## 8. Conclusions

The immature HPO axis in pubertal animals may lead to defects in follicular signaling and steroidogenesis. The estradiol and LH concentrations in the follicular fluid of pubertal female animals are approximately half of those in adult female animals, so AFF technology provides a more suitable follicular environment for pubertal oocytes. However, the reproducibility and efficiency of AFF still need to be improved. The factors affecting the success rate of AFF may be multifactorial. Establishing more standardized procedures for different animal species is necessary to ensure their efficiency, including transferring appropriate numbers of COCs and using appropriately sized needles, among other things. Simplifying surgical procedures, transitioning to minimally invasive methods, and enhancing the proficiency of professionals can reduce the physical damage to animals, oocytes, and embryos. In addition, transferring immature COCs appears to be more efficient than transferring in vitro matured COCs. Therefore, improving the synchronization of donor COCs and recipient follicular maturation may be one of the future research directions. AFF technology does not rely on complex laboratory equipment, and this innovative method offers a new approach for embryo production, cryopreservation, and transgenic technology.

## Figures and Tables

**Figure 1 vetsci-11-00276-f001:**
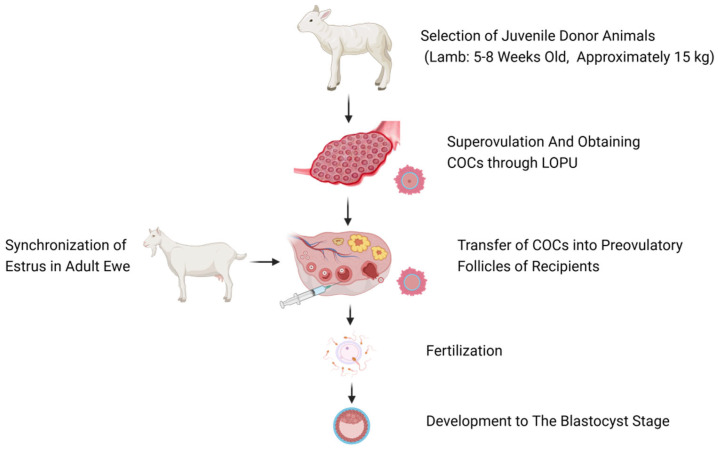
Overview of AFF in ewes (Figure created with BioRender.com, accessed on 6 May 2024).

## Data Availability

No new data were created or analyzed in this study. Data sharing is not applicable to this Review.

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
