# Peer review of "Allogenic Follicular Fosterage Technology: Problems, Progress and Potential"

_vetsci, 2024, doi:10.3390/vetsci11060276_

Round 1

Reviewer 1 Report

Comments and Suggestions for Authors

The review is very current and important in the reproduction area. I like the manuscript as it is, but I have a few stylistic recommendations: in vitro and in vivo should be written in italics in the whole document. Line 72 - transferring (t is missing); line 286 - add the number of citation for Martinez et al; line 293 - The number (T is missing); line 316 - put space between the numbers and mm.  

Author Response

I sincerely appreciate your recognition! Regarding your suggestions, I have made the necessary revisions. (Please understand that the number of lines has changed due to the adjustment of the article.)

1. I have italicized the terms "in vitro" and "in vivo" throughout the entire article.
2. Line 201 has been corrected to include the missing "T".
3. Line 289 has been updated to include the citation number for Martinez et al.
4. Line 295 has been revised to add the missing "T".
5. Line 320 has been edited to ensure there is a space between the number and "mm" for readability.

I hope this updated translation meets your satisfaction. Please let me know if you have any further comments or suggestions. Wish you a happy life and smooth work!

Reviewer 2 Report

Comments and Suggestions for Authors

The article is written concisely and clearly. The subchapters create a logical sequence, allowing for a better understanding of the technically difficult topic that the authors have undertaken.

 In the first chapters, especially in 2. "History of JIVETand AFF" and 3. The Impact of AFF on Oocyte Maturation and Embryo Development", one may get the impression that the author is not objective and creats the thesis about the superiority of AFF technology over embryo in vitro culture (in context of oocyte and embryo development). In the text you can find many citations and described works whose results indicate a higher level of success after using IVEP vs. AFF. However, the message of the entire content of the paper leads the reader to the conclusion that this is a matter of further research, suggesting, in the reviewer's opinion, the superiority of the AFF method.

In the reviewer’s opinion some small corrections could transform this good text to more objective form.

 The aspect of the number of oocytes transferred to the preovulatory follicle seems to be unexplained. In all pointed species the preovulatory follicule may occur in numer of one, two or even more. In the text we can find the phrase:

„Sprícigo and colleagues have injected up to 25 COCs into preovulatory follicles in cattle [47] (line 300).  Whether this should be understood as transfer of 25 oocytes to one follicle or 25 oocytes to the follicles of one cow?. It seems to be very interesting and technicaly important subject. This should be specified.

 While reading the paper the question also arises what about the recipient's oocyte from the preovulatory follicle. Is it assumed that it will remain in the pool of embryos obtained by AFF technic? What is the solution to differentiate donor’s and recipient’s origin material?

 Typo correction:

Line 192: word „receptors” seams to be mistaken with recipients?

Line 293. : the lack of „T”

 The paper is highly interesting and presents the subject in a multi-genre manner, which is not available in the currently available literature. In my opinion should be published after minor correction, which were presented above.

Author Response

Thank you for your suggestions on this article, which are very meaningful for us to improve this review. I have made modifications to the article based on your suggestions. Here are my responses to your questions and suggestions:

  1. Due to the lack of extensive research on AFF and the significant differences in results obtained from different studies, it is currently difficult to conclude that IVEP is superior to AFF or vice versa. The design and specific details of AFF studies vary, which may affect the final success rate. Therefore, further research is needed to determine whether the success rate of optimized AFF procedures is superior or inferior to that of IVEP. For example, it has been found in studies that the success rate of transferring immature COCs is higher than that of transferring mature COCs, and the success rate of transferring COCs with more layers of granulosa cells is higher than that of COCs with fewer layers of granulosa cells. Therefore, improving the synchronization of COCs and recipient follicle maturation, as well as reducing physical damage to COCs during follicular aspiration, may increase the final success rate of AFF.
  2. " Whether this should be understood as transfer of 25 oocytes to one follicle or 25 oocytes to the follicles of one cow? " Regarding this question, we have clarified in "5.2. Number and Quality of Transplanted Oocytes" of the article that it is transferring 25 oocytes into one dominant follicle.
  3. "What is the solution to differentiate donor's and recipient's origin material?" We have addressed this question in the "3. AFF Steps and Distinguishing AFF COCs from Originals" section of the article. Specifically, it states: "In the study conducted by Andino and colleagues, they differentiated the native COC from the AFF COCs based on the characteristic large, yellow, mucoid expanded cumulus cells of the latter. AFF-derived COCs had smaller and less cellular cumulus masses because most of their cumulus cells were stripped during the initial recovery from their native follicles. When the cumulus cells of AFF-derived COCs expanded, they appeared to have a clear rather than yellow intercellular matrix and were less mucoid in appearance [11]. To further accurately determine the origin of the embryos, the recovered embryos were carefully bisected, and the resulting cells were then submitted for parentage testing [11]. The offspring obtained through AFF can be confirmed using genotype analysis [13]."
  4. Line 186: Change the word "receptors" to "recipients".
  5. Line 301: Add the missing "T".

Thank you for your support! I wish you a happy life and smooth work!

Reviewer 3 Report

Comments and Suggestions for Authors

The idea and the subject of this review is interesting, it can be promising from both biological and practical viewpoints. Nevertheless, the form of the review requires substantial re-organization, updating and improvement.

ABSTRACT contains a long introduction describing the significance of JIVET and AFF, but only the last sentence concerns the review itself. I would suggest to reduce general part and to expand the Abstract with the concrete conclusions.

INTRODUCTION describes history and efficiency of JIVET, which should be presented in the next chapters (“History of JIVET and AFF” and „The Impact of AFF on Oocyte Maturation and Embryo Development“), but it does not demonstrate the requirement of such review (whether such reviews have been issued, if yes, why the present review is necessary).

The contents of “History of JIVET and AFF” and „The Impact of AFF on Oocyte Maturation and Embryo Development“ is similar. Therefore, I would suggest to merge these chapters.

The review is focused mainly to technical problems of JIVET and AFF, but the causes of these problems could be explained by biological data concerning differences in juvenile and adult follicles, species-specific differences in JIVET and AFF efficiency, age- and species-specific differences production and transport of their nutrients and signaling molecules affecting cytoplasmic maturation of oocytes, embryogenesis, their cell cycle, apoptosis, viability, differentiation, development etc. Such biological aspects should represent the major part of such review because only their understanding could explain and improve the efficiency of JIVET and AFF.

The title “Present Problem“ is inadequate, and this chapter states problems, but not their explanation and solution.

„Summary and Prospects“, like other chapters represents rather the list of the available data, but not its critical discussion, the biological concept, explaining the current problems and outlining their solutions. Therefore, the review summarizes the current state of art of methodology of JIVET and AFF, but it did not explain their biological bases, solution of the current problems in their application and the direction of the study towards these problems. I would suggest to recruit for generation this paper the good specialist in reproductive biology, which could deepen the biological aspect of this review.

Author Response

Thank you immensely for your thorough guidance and invaluable suggestions, which have significantly enhanced the quality of this article. Following your directions, we have thoroughly revised the article, focusing on the key points you mentioned:

  1. We have streamlined the Abstract by reducing the general introduction and expanding it with more concrete conclusions.
  2. We moved the section describing the history and efficiency of JIVET in the Introduction to the subsequent section "2. The History of JIVET and AFF and Their Impact on Oocyte and Embryonic Development". We also added a statement in the Introduction explaining the necessity for such a review.
  3. We have merged the contents of the two chapters, "The History of JIVET and AFF" and "The Impact of AFF on Oocyte Maturation and Embryo Development" into the chapter titled " The History of JIVET and AFF and Their Impact on Oocyte and Embryonic Development" and revised the content within it.
  4. We have added a new section titled "6. Factors Influencing Oocyte Developmental Differences Between Prepubertal and Adult Female Animals," which includes subsections “1. Follicular Microenvironment”, “6.2. HPO Axis and Follicular Diameter”, and “6.3. Interaction between Oocytes and Granulosa Cells”.
  5. We have renamed the title "Present Problem" to "7. Potential Negative Factors Influencing the Success Rate of AFF" and revised the content within it.
  6. We have renamed the title "Summary and Prospect" to "8. Conclusion" and revised the content within it.

Thank you for your suggestions. I wish you a happy life and smooth work!

Round 2

Reviewer 3 Report

Comments and Suggestions for Authors

The MS has been improved